# The Role of Knowledge, Attitude, Confidence, and Sociodemographic Factors in COVID-19 Vaccination Adherence among Adolescents in Indonesia: A Nationwide Survey

**DOI:** 10.3390/vaccines10091489

**Published:** 2022-09-07

**Authors:** Defi Efendi, Sabira Ridha Rifani, Ariesta Milanti, Ferry Efendi, Cho Lee Wong, Yeni Rustina, Dessie Wanda, Dian Sari, Ivonne Junita Fabanjo, Elzina Dina De Fretes, Rini Wahyuni Mohamad, Oktoviandi Sawasemariay, Ruth Harriet Faidiban, Qoriah Nur, Indah Benita Tiwery, Mega Hasanul Huda, Oktovina Mobalen

**Affiliations:** 1Department of Pediatric Nursing, Faculty of Nursing, Universitas Indonesia, Depok 16424, Indonesia; 2Neonatal Intensive Care Unit, Universitas Indonesia Hospital, Depok 16424, Indonesia; 3Undergraduate Program, Faculty of Nursing, Universitas Indonesia—Jl. Prof. Bahder Djohan, UI Campus, Depok 16424, Indonesia; 4The Nethersole School of Nursing, The Chinese University of Hong Kong, Shatin, New Territories, Hong Kong SAR, China; 5Faculty of Nursing, Universitas Airlangga—Jl. Dr. Ir. H. Soekarno, Mulyorejo, Kec. Mulyorejo, Kota SBY 60115, Indonesia; 6Undergraduate Nursing Study Program, Prima Nusantara Institute of Health—Jl. Kusuma Bhakti, No. 99, Bukittinggi 26113, Indonesia; 7Manokwari Diploma III Nursing Study Programme, Poltekkes Kemenkes Sorong, Jl. Slamet Riyadi Kampung Ambon Atas, Manokwari 98311, Indonesia; 8Fakfak Diploma III Nursing Study Program, Poltekkes Kemenkes Sorong, Jl. Diponegoro No.1 Fakfak, Fakfak 98011, Indonesia; 9Pediatric Nursing Department, Faculty of Sport and Health, Universitas Negeri Gorontalo—Jl. Jend. Sudirman Dulalowo, UNG Campus, Dulalowo 96128, Indonesia; 10Pediatric Nursing Department, Polytechnic of Health, Ministry of Health, Jl. Padang Bulan II Kel. Hedam Kec. Heram, Kota Jayapura 99351, Indonesia; 11Nursing Science Study Program, Faculty of Health, Universitas Kristen Indonesia Maluku, Indonesia-Jl. OT. Pattimaipauw, Tanah Lapang Kecil, Ambon City 97115, Indonesia; 12Research and Development Center Hermina Hospital Group, Jakarta. Hermina Tower, Jl. Selangit B-10 Kavling 4, Kemayoran 10620, Indonesia; 13Diploma IV Nursing Study Program, Poltekkes Kemenkes Sorong, Jl. Basuki Rahmat Km 11.5, Sorong 98417, Indonesia; 14Pediatric Nursing Care Unit, Sulianti Saroso Hospital, Jakarta 14340, Indonesia

**Keywords:** adherence, adolescents, attitude, confidence, COVID-19, knowledge, sociodemographic factors, vaccination

## Abstract

COVID-19 vaccination in adolescents is important because the adolescent population has the highest incidence of COVID-19. This study aimed to explore the factors associated with COVID-19 vaccination adherence among Indonesian adolescents. This cross-sectional study involved 7986 adolescents, polled through online and offline surveys conducted in six major islands of Indonesia. The online questionnaire was distributed through popular social messaging and social media platforms. Our team also contacted schools and public places to recruit participants from remote areas. In total, 7299 respondents completed the questionnaire. Binary logistic analysis revealed that higher levels of knowledge, positive attitudes, and confidence in the COVID-19 vaccine were significantly associated with higher COVID-19 vaccination adherence in adolescents. Sociodemographic factors were also significantly associated with higher adherence to vaccination programs. Meanwhile, younger age and habitation in private housing were related to lower adherence to the vaccination program. Parental factors related to adolescent compliance were education level, household income, history of infection of family or friends with COVID-19, and working status. The national authorities and stakeholders should take extensive measures to increase attitude, knowledge, confidence, and family support among adolescence through multiple channels.

## 1. Introduction

Vaccination against coronavirus disease 2019 (COVID-19) is crucial for achieving herd immunity [1]. COVID-19 vaccination in adolescents is important because they exhibit the highest incidence of COVID-19 [2], which plays an essential role in COVID-19 transmission in the community [3]. According to CDC data, approximately 1700 per 100,000 adolescents aged 12–17 years were infected with COVID-19 in the United States as of 15 January 2022 [2]. COVID-19 can cause severe and prolonged symptoms in adolescents, particularly those who are unvaccinated [4].

In Indonesia, the incidence of COVID-19 cases are overwhelmingly within the adolescent and productive populations, with teenagers comprising approximately 10.4% of the total cases reported [5]. The national coverage of people who received at least two doses of the vaccine was 72.63%. Meanwhile, coverage among the 12–17 years age group was 82.6%, 52.51% lower than that in the healthcare group. Although the vaccination rate of adolescents is higher than the national rate, the need to boost their vaccination coverage to 100% persists as the adolescent population is considered to have a potent risk of being a SARS-CoV-2 reservoir which may lead to community epidemic outbreaks [6,7]. It has been reported that a vaccinated household can reduce symptomatic cases by 50% compared to an unvaccinated one [8,9]. Thus, there is a pressing need to examine the factors influencing adolescent vaccination to improve vaccination coverage.

A scoping review by Liu, Ma, Liu, and Guo [10] found that the median vaccination rate among adolescents was 50.40%. The majority of the studies showed that adolescents’ reluctance to get vaccinated was mostly due to concerns about vaccine safety, the effectiveness of vaccines, and potential side effects. In line with this finding, Cai et al. [11] suggested that adolescents’ reception of the COVID-19 vaccine could be influenced by their level of education and confidence in its safety and efficacy. Hence, adolescents’ knowledge of the COVID-19 vaccine must be investigated to determine the progress of their understanding and willingness to receive it.

Most adolescent vaccination research publications have focused on adolescents’ attitudes and willingness to receive the COVID-19 vaccine [10]. Thai youth have a negative attitude toward the COVID-19 vaccine due to concerns about the vaccine’s side effects [12]. Moreover, adolescents in Hong Kong [13], Canada [3], and the United States [14] also reported major concerns regarding vaccine safety and efficacy. Another study in China found that 76.3% of adolescents thought the COVID-19 vaccine was safe, and 75.59% of Chinese adolescents would receive it [11]. Contrastingly, a study of adolescents in the United Kingdom found that 50.1% had scheduled vaccination against COVID-19, 37.0% were hesitant, and 12.9% decided not to be vaccinated [15].

Previous research on COVID-19 vaccination in adolescents shows that this topic still lacks attention [10]. While there have been several studies regarding adolescents’ attitudes and willingness to accept the COVID-19 vaccine, to the best of our knowledge, there has not been any research on adolescents’ adherence to COVID-19 vaccination. In Indonesia, no research data regarding adolescent vaccination exist, particularly regarding adolescents’ adherence to COVID-19 vaccination. It is critical to conduct research on the factors that influence adolescents’ willingness toward and adherence to vaccination to develop strategies that can help accelerate COVID-19 vaccination implementation. Thus, this study aimed to investigate whether the levels of knowledge, attitude, trust, and confidence are determining factors for adolescents’ adherence to the COVID-19 vaccine. It was hypothesized that adolescents’ levels of knowledge, attitude, trust, and confidence were associated with their adherence to the COVID-19 vaccine.

## 2. Materials and Methods

### 2.1. Study Setting, Design, Participants, and Sampling

This nationwide study was conducted in Indonesia. Data were collected from six major islands of Indonesia: Java, Sumatra, Sulawesi, Borneo, Bali/East Nusa Tenggara/West Nusa Tenggara, and Maluku/Papua [16]. The adolescent population in Indonesia was approximately 46 million at the time of this study, comprising 17% of the total population [16].

This study had two phases: translation and validation of the questionnaires, and a cross-sectional analytical study. In both phases, we examined Indonesian adolescents aged 12–17 years who could read and write in Bahasa Indonesia. Adolescents who had not received the first or second dose for the following reasons were excluded from this study: (i) a shortage of vaccines, (ii) contraindications to the vaccine, (iii) vaccination scheduled for a later date, and (iv) did not enclose reasons for not getting vaccinated.

The first phase was conducted between February and March 2022. We recruited 556 participants and conducted a follow-up for reliability testing. The Cronbach’s alpha values for knowledge, attitudes, and beliefs about COVID-19 were 0.669, 0.710, and 0.932, respectively. The sample size was calculated based on the number of question items for the factor analysis (a minimum of 10 participants for each item) [17]. The second phase was conducted between March and June 2022. The calculation of the sample size was based on the following criteria from a previous study [18]: (1) 80% power of the study, (2) the number of independent variables (*n* = 20), (3) alpha = 0.05, and (4) effect size = 0.1. A sample size of 1523 participants was estimated to be adequate to achieve acceptable external validity in evaluating the research outcomes.

### 2.2. Measures

Measures from previous studies were adopted to assess the dependent and independent variables in this study [11,19,20]. There are four sections with 31 items that measure knowledge, attitude, confidence, and adherence to the COVID-19 vaccination.

Knowledge of the COVID-19 vaccine is a 10-item questionnaire from Mohamed et al. [20]. Knowledge is measured through “yes,” “no,” and “do not know” responses. The total score is 10, and a mean score of 3.9 is used as a cutoff to categorize those with good knowledge and those with poor knowledge. Attitudes toward the COVID-19 vaccine were measured using a six-item questionnaire adopted from Cai et al. [11]. A median of 11 was used as a cut-off for categorization (positive attitude < 11 and negative attitude > 11). In the third section, a 14-item questionnaire, originally developed by Freeman et al. as the Oxford COVID-19 vaccine confidence scale, is used to assess confidence in the COVID-19 vaccine [19]. The scale is coded from 1 to 5, with “do not know” answers scored as 0. We used the median value as the cut-off point, where <median is classified as high confidence and> median is classified as low confidence. The details of these instruments are provided in Appendix A. Finally, a structured questionnaire was developed to assess adolescent adherence to COVID-19 vaccination. COVID-19 vaccine adherence was defined as complete vaccination with two obligatory vaccine doses received [21]. The reasons why adolescents chose to not be vaccinated were collected as determining criteria for the targeted sample within our sample. Those who were unvaccinated for nonmedical and technical reasons (e.g., perceptions related to the vaccine and its effects and parental prohibition) were regarded as the nonadherent group, as mentioned in Figure 1.

The questionnaire began with sociodemographic characteristics that were classified from previous studies, including age [22,23], sex, residential area [24], distance between home to vaccination sites [25], health status [11], COVID-19 status, chronic disease status, and congenital disease status [12,20,26], and media used to access COVID-19 vaccine information [27]. Parents’ education and occupation, household income [28,29], financial problems during the pandemic [3], and history of family members or friends infected with COVID-19 [20] were also included in this section.

#### Translation Process

We obtained permission from the original authors via e-mail to adapt their original instruments for the Indonesian population. The six-step translation and validation of the instrument, as outlined by Sousa, was used in this phase [30]. In the first step, a pair of translators whose mother language was Bahasa Indonesia translated the questionnaires (Target Languages [TL] 1 and 2). Both translators were knowledgeable about health terminology and Indonesian cultural nuances. Next, the TL 1 and TL 2 versions were compared using a third independent translator. The ambiguities were resolved to reach the preliminary initial translated version (PI-TL), which was then translated back into English by two native English translators. The back-translated versions were then compared and synthesized into the pre-final version of the instrument (P-FTL). In step five, we conducted a pilot test of the P-FTL in an Indonesian sample. Twenty participants evaluated the instructions, items, and response format clarity. Finally, full statistical testing was performed on 556 participants. The items in the final version of the translated questionnaire were revised and refined based on the test.

### 2.3. Data Collection and Analyses

We distributed the online questionnaire via the online platforms WhatsApp, Line, Twitter, and Instagram, as these are the most popular social media platforms among the younger generation in Indonesia [31]. In remote provinces, our team contacted schools and public places to recruit participants. We informed the adolescents about the study, and those who were willing to participate completed the paper-based questionnaire.

All variables were summarized using descriptive statistics as appropriate. Bivariate analysis, especially the chi-square test, was conducted to examine the relationship between two categorical variables. Binary logistic regression was conducted to predict the factors associated with COVID-19 vaccination adherence among Indonesian adolescents. The output was categorized into dichotomous dependent variables based on independent variables, which can be either continuous or categorical. All data were analyzed using SPSS version 25 (IBM Corp., Armonk, NY, USA).

### 2.4. Ethical Considerations

This study was reviewed and approved by the Institutional Review Board of the Faculty of Nursing Universitas Indonesia (approval number Ket-83/UN2. F12. D1.2.1/PPM.00.02/2022). Participation was voluntary, and all respondents were provided with information about the research. This study complied with the ethical principles of the Declaration of Helsinki.

## 3. Results

A total of 7986 adolescents were recruited from six major islands of Indonesia. Of these, 383 did not allow their data to be published for research purposes, and 304 submitted incomplete questionnaires, which resulted in their exclusion from the study. Finally, 7299 participants were included in the study. The sociodemographic characteristics of the participants are summarized in Table 1.

Approximately half of the participants (53.3%, *n* = 3886) were willing to get vaccinated. Fifty-five percent (55.1%, *n* = 4021) of the participants did not adhere to vaccination. The primary reason the participants did not adhere to the vaccination schedule was that their parents forbade them (49.5%, *n* = 1989). The majority of the participants were from the Maluku and Papua islands (63.2%, *n* = 5613), were in the middle adolescent group (63%, *n* = 4597), and were female (63%, *n* = 4597). Most of the participants resided in urban areas (60.2%, *n* = 4391). Approximately half of the participants (50.4%, *n* = 3680) lived within four kilometers of the COVID-19 vaccination centers. Most participants (75.5%, *n* = 5514) lived in private homes. Regarding health status, the majority of participants (76.2%, *n* = 5559) reported good health status, 77.4% (*n* = 5646) had never been infected with COVID-19, 95.6% (*n* = 6967) had no chronic diseases, and 95.4% (*n* = 6969) had no congenital diseases. Most participants (72.6%, *n* = 5300) accessed COVID-19 information through electronic media. Approximately half of the adolescents had higher levels of knowledge (54.6%) and positive attitudes (54.6%) toward COVID-19 vaccinations. Nevertheless, 50.5% (*n* = 3684) had low levels of trust in COVID-19 vaccination.

Information on the parents of the participants was also obtained. Forty-two percent (*n* = 3110) of parents attended senior high school. The majority of parents (83.4%) were employed, and 92.5% of parents earned less than IDR 5,100,000 per month. All parents had experienced financial difficulties during the pandemic.

Table 2 shows that the variables that have significant relationships with COVID-19 vaccination adherence are willingness to get vaccinated, island, age, gender, area of residence, distance from house to the vaccination site, housing status, health status, COVID-19 status, chronic disease status, congenital disease status, and source of information regarding COVID-19 vaccination. Adherence to COVID-19 vaccination was also found to be significantly related to parents’ education level, household income, history of family or friends getting infected with COVID-19, parents’ occupation, parents’ education level, and adolescents’ attitude toward and confidence in the COVID-19 vaccine.

In the multivariate analysis, the associations between independent and dependent variables were assessed using logistic regression. Table 3 shows that the adolescents who were not willing to get vaccinated had an adherence rate [AOR = 0.159; 95% CI = 0.130-0.195] 0.159 times lower than vaccinated adolescents. The place in which the adolescents lived was also a determinant of vaccine compliance. Adolescents residing in Java Island reported vaccine adherence 1.994 times higher [AOR = 1.994; 95% CI = 1.600–2.485] than those residing in Maluku and Papua Islands.

Based on the adolescent factors, age [AOR = 0.498; 95% CI = 0.417–0.595], area of residence [AOR = 2.007; 95% CI = 1.701–2.368], housing status [AOR = 0.555; 95% CI = 0.460–0.670], and health status [AOR = 3.273; 95% CI = 1.651–6.488] were factors associated with adherence to the COVID-19 vaccine. Early adolescents aged 12–14 years were 0.498 times less adherent than middle adolescents (15–17 years). Adolescents who lived in urban areas were 2.007 times more likely to adhere to COVID-19 vaccination than those who lived in rural areas, whereas those who lived in private houses were 0.555 times less likely to adhere to COVID-19 vaccination than those who lived in rented houses. Adolescents with good health status showed 3.273 times higher adherence to COVID-19 vaccination than those with poor health status.

Several parental factors also influenced adolescents’ adherence to vaccines, namely education level [AOR = 0.396; 95% CI = 0.312–0.502], household income [AOR = 0.722; 95% CI = 0.532–0.979], and employment status [AOR = 1.287; 95% CI = 1.040–1592]. The history of a family member or friend being infected with COVID-19 [AOR = 1.345; 95% CI = 1.116–1.622] also influenced adherence. Adolescents with parents who had a junior high school education or lower tended to comply with the COVID-19 vaccination 0.396 times less than those whose parents had a college education. Adolescents with a household income of less than IDR 5,100,000 were 0.722 times less likely to adhere to COVID-19 vaccination than those with household incomes over IDR 5,100,000. Adolescents with working parents were 1.287 times more likely to adhere to the COVID-19 vaccination than those whose parents did not work. Meanwhile, adolescents with a history of family members or friends infected with COVID-19 were 1.345 times more likely to be compliant than those who did not. Other factors associated with adolescent vaccine adherence included a high level of knowledge [AOR = 1.962; 95% CI = 1.568–2.455], a positive attitude [AOR = 7.072; 95% CI = 5.652–8.849], and a high level of trust in the COVID-19 vaccine [AOR = 1.872; 95% CI = 1.495–2.343].

Figure 1 illustrates the percentages of the reasons for COVID-19 vaccine nonadherence. Most participants (57.5%) reported that they did not complete two doses of vaccination because they were forbidden by their parents. Nearly half of the participants mentioned this as the reason they did not receive the first dose of the COVID-19 vaccine.

Figure 2 depicts the proportion of information sources for the COVID-19 vaccine. The majority of the participants primarily obtained their information from social media and online media. Both groups of participants who adhered or did not adhere to the COVID-19 vaccination program accessed social media and online media for COVID-19 vaccine information.

## 4. Discussion

This study was the first nationally representative study with a large sample size to investigate COVID-19 vaccination adherence among Indonesian adolescents, as well as its determining factors. This study found that over half of the participants did not adhere to the COVID-19 vaccination program. This study identified several key factors related to adolescents’ adherence to the basic COVID-19 vaccination program. The study revealed that residential areas (Java Island and urban areas), good health status, higher knowledge and vaccine confidence, and positive attitudes were related to higher adolescent compliance with the COVID-19 vaccination program. Conversely, younger age (12–14 years) and residence in a private house correlated with lower adolescent adherence to COVID-19 vaccination. Additionally, adolescents with employed parents were more likely to be vaccinated. Meanwhile, lower parental education and income may decrease adolescents’ COVID-19 vaccine compliance.

Sociodemographic factors were among the factors that influenced adolescents’ adherence to the COVID-19 vaccination. This study found that the adherence level of adolescents living in urban areas was higher than that of those residing in rural areas. This finding is in line with previous findings, which found that vaccination willingness [32] and coverage of children and adult populations were lower in rural areas than in cities [33]. This might be due to access to healthcare facilities, which are comparably lower in rural areas [33]. Additionally, city dwellers commonly have better access to information about the effectiveness of the vaccine, resulting in lower vaccine hesitancy [34].

Before the pandemic, people living in rural areas tended to report that they did not have enough healthcare providers or facilities compared with those living in cities [35]. During the pandemic, access to COVID-19 vaccination in villages was limited [33]. In Indonesia, rural areas and outside Java Island tend to have a lower vaccine uptake. However, regional differences in vaccination rates are not related to regional differences in vaccination status [5]. The availability of healthcare facilities and providers, which vary between different areas of Indonesia, might contribute to different vaccination rates [36]. Apart from the access to and availability of health care resources, villagers and city dwellers have differing views on the seriousness of COVID-19 infection [33,37]. City residents took COVID-19 more seriously and were more compliant with COVID-19 preventive measures [33,37]. Additionally, parents in rural areas were three times more likely to report that they “definitely will not” get their children vaccinated compared to parents living in cities [33]. This study also found that the main reason adolescents did not receive the COVID-19 vaccine was parental prohibition.

Regarding knowledge level, adolescents with higher levels of knowledge were more likely to adhere to COVID-19 vaccination than those with lower levels of knowledge. A lack of knowledge about COVID-19 vaccination might lead to misperceptions about the vaccine. This finding aligns with the results of previous studies, which suggest that knowledge of the disease and the corresponding vaccine affects the intention to get vaccinated [38]. In addition, a study on the HPV vaccine found that individuals who lacked knowledge about HPV and its vaccine were more likely to be unvaccinated than those who had better knowledge [39]. Rehati et al. [40] also suggested that vaccine hesitancy among Chinese students was linked to limited health literacy, lower risk awareness, and lower exposure to accurate information about the importance of vaccination. Obtaining more information about the safety and efficacy of the COVID-19 vaccine, as well as vaccination requirements of schools, were the most frequently reported factors that could increase the vaccination intention of adolescents [14].

Furthermore, positive attitudes toward the COVID-19 vaccine were found to be significantly related to COVID-19 vaccination adherence. Similarly, a previous study found that the majority of their participants had positive attitudes and reported their intention to get COVID-19 vaccinations [11,41]. Adolescents with positive attitudes toward the vaccine believe that it is safe and has minimal side effects; thus, they are more willing to accept it. A study by Cvjetkovic, Jeremic, and Tiosavljevic [42], which adopted the theory of planned behavior, concluded that positive attitudes could lead individuals to vaccinate themselves.

Confidence in the COVID-19 vaccine was also associated with COVID-19 vaccination adherence. This was confirmed by a previous study that concluded that 39% of adolescents who had been vaccinated were influenced by their confidence in the vaccine and perception of increased disease risk if they were not vaccinated [13]. In the present study, 22.9% of adolescents did not adhere to the COVID-19 vaccination program due to fears concerning the vaccine’s side effects, while 15.1% did not adhere to it because of their hesitancy regarding vaccine effectiveness. This evidence strengthens the notion that confidence in the vaccine is highly related to the decision to get vaccinated.

The major factor that leads to adolescents’ willingness to get vaccinated is information about the safety and efficacy of the COVID-19 vaccine. Information about the vaccine, which was provided by authorities or health care professionals, may affect confidence in, acceptance of, and uptake of vaccines among adolescents [14]. More confidence in the effectiveness of the COVID-19 vaccine might lead to fewer adolescents refusing vaccination [40]. Public skepticism about the vaccine and worries about potential side effects were barriers to achieving herd immunity for COVID-19 [13,43]. It is important to provide accurate information about COVID-19 vaccine efficacy and safety [44] to counter misinformation about the vaccine’s side effects and effectiveness, as well as the beliefs of antivaccination groups [3]. Individuals might have different levels of confidence in the COVID-19 vaccine due to a lack of information about the vaccine type, availability, and safety profile [45].

In the present study, we also found that the main reason adolescents were not vaccinated was parental objection, which was as high as 49.56%. Although we did not specify the reason for ‘parental prohibition’ by age group, we found that the nonadherence rate of early adolescents was higher than that of middle adolescents (75.4% and 47.4%, respectively). This might be due to early adolescents’ lack of autonomy in making their own health decisions compared to middle adolescents [46,47]. A study of adolescents aged 16–18 years in Israel showed that their level of involvement in the vaccination decision making was significantly associated with actual vaccination [48]. Middle adolescents are more likely to make their own decisions regarding their health [48]. Moreover, a study in the US indicated that even adolescents aged 12–13 years already participated in vaccination decision making [49]. In societies such as industrialized Western countries, where autonomy, personal success, and self-direction are emphasized, parents are more likely to involve their children in decision-making [50]. However, in Indonesia, children are obliged to obey their parents even after they reach adulthood [50,51].

## 5. Limitations

This study has some limitations. First, careless responses may have affected the results, as this study was partially conducted online [52]. However, the large sample size of this study might compensate for the minor mistakes of some participants, who may have been careless in filling out the questionnaire [52]. Second, Indonesia is a large archipelagic country, in which accessibility remains a challenge. Our study was conducted in six major islands of Indonesia, which might have resulted in an overestimation of COVID-19 vaccine compliance in the adolescent population. However, this investigation was the largest study on COVID-19 vaccination among adolescents and involved numerous data collectors from 19 different sites in Indonesia. This may have attenuated the issue of data equity. Furthermore, paper-based questionnaires were used to facilitate responses from remote and rural areas, where online questionnaires were not applicable.

## 6. Conclusions

Less than half of the participants in the present study complied with the COVID-19 vaccination program. A positive attitude toward the vaccine was the primary factor influencing adolescents’ compliance with COVID-19 vaccination. Levels of knowledge and confidence regarding the COVID-19 vaccine were also associated with COVID-19 vaccination adherence. Meanwhile, parental education level and income were key factors associated with adolescents’ adherence to COVID-19 vaccination. Adolescents who did not receive COVID-19 vaccination reported that the main reason was parental prohibition and lack of belief in the vaccine. These findings highlight the urgency of aggressive measures to increase adolescent adherence to COVID-19 vaccination, especially for families of lower socioeconomic status. It is important to educate parents and involve them in juvenile COVID-19 vaccination to maximize vaccination coverage in Indonesia. Additionally, a special approach is needed to reach adolescents who are unwilling to get vaccinated against COVID-19.

## Figures and Tables

**Figure 1 vaccines-10-01489-f001:**
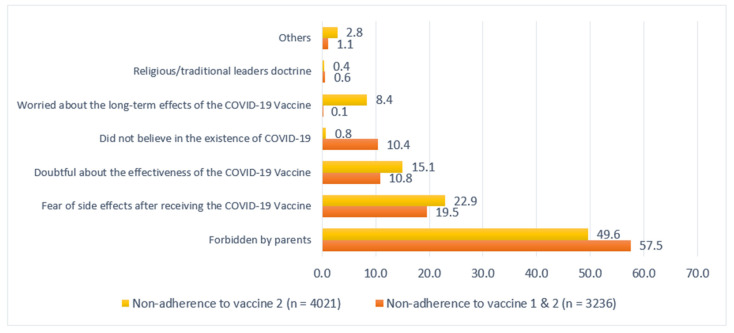
The reasons for vaccine non-adherence among participants.

**Figure 2 vaccines-10-01489-f002:**
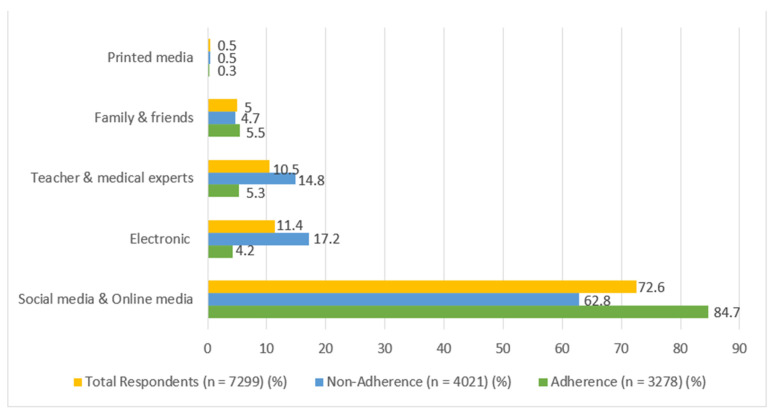
COVID-19 Vaccine Information Sources.

**Table 1 vaccines-10-01489-t001:** Sociodemographic characteristics (*n* = 7299).

Variable	*n*	%
Vaccination Willingness	2549	34.9
Unwilling	864	11.8
Doubtful	3886	53.3
Ready		
COVID-19 Vaccination Adherence	3278	44.9
Yes	4021	55.1
No		
Island		
Java	1474	20.2
Sumatra	179	2.4
Sulawesi	863	11.8
Borneo	114	1.6
Bali/East and West Nusa Tenggara	56	0.8
Maluku and Papua	4613	63.2
Age		
Early adolescent (12–14 years)	2014	27.6
Middle adolescent (15–17 years)	5285	72.4
Sex		
Male	2702	37
Female	4597	63
Residential Area	4391	60.2
Urban	2908	39.8
Rural		
Distance from House to COVID-19 Vaccination Site	3680	50.4
<4 km	3619	49.6
>4 km		
Housing Status	5514	75.5
Private housing	1785	24.5
Rented housing		
Health Status	5559	76.2
Good	1629	22.3
Fairly good	111	1.5
Poor		
COVID-19 Status	1653	22.6
Prior COVID-19 infection	5646	77.4
No prior COVID-19 infection		
Chronic Disease Status		
Present	323	4.4
Absent	6976	95.6
Congenital Disease Status	339	4.6
Present	6960	95.4
Absent		
COVID-19 Vaccine Information Resources	5300	72.6
Electronic	1999	27.4
Non-electronic		
Parents’ Education Level	634	8.7
Elementary School	1262	17.3
Junior High School	3110	42.6
Senior High School	2293	31.4
College		
Household Income	6751	92.5
<IDR 5,100,000	548	7.5
≥5,100,000		
Financial Problems (During the COVID-19 Pandemic)	7299	100
Yes	0	0
No		
History of Family Members or Friends Infected with COVID-19	2013	27.6
Yes	5286	72.4
No		
Parents’ Employment Status	6087	83.4
Employed	1212	16.6
Not employed		
Adolescents’ Knowledge Level of COVID-19 Vaccine	3712	50.9
High	3587	49.1
Low		
Attitude Towards COVID-19 Vaccine	3986	54.6
Positive	3313	45.4
Negative		
Confidence in COVID-19 Vaccine	3615	49.5
High	3684	50.5
Low		
Total	7299	100

**Table 2 vaccines-10-01489-t002:** Bivariate analysis using chi-square test (*n* = 7299).

Variables	Adherence				χ^2^	
	Yes		No			*p*-Value
	*n*	%	*n*	%		
Willing to Get Vaccinated					2261.324 ***	0
Unwilling	198	7.8	2351	92.2		
Doubtful	444	51.4	420	48.6		
Ready	2636	67.8	1250	32.2		
Island					1681.159 ***	0
Java	1267	86	207	14		
Sumatra	137	76.5	42	23.5		
Sulawesi	381	44.1	482	55.9		
Borneo	99	86.8	15	13.2		
Bali/East and West Nusa Tenggara	53	94.6	3	5.4		
Maluku and Papua	1341	29.1	3272	70.9		
Age					462.499 ***	0
Early adolescent (12–14 years)	496	24.6	1518	75.4		
Middle adolescent (15–17 years)	2782	52.6	2503	47.4		
Sex					159.911 ***	0
Male	954	35.3	1748	64.7		
Female	2324	50.6	2273	49.4		
Residential Area					671.199 ***	0
Urban	2511	57.2	1880	42.8		
Rural	767	26.4	2141	73.6		
Distance From House to COVID-19 Vaccination Site					37.611 ***	0
<4 km						
>4 km	1783	48.5	1897	51.5		
	1495	41.3	2124	58.7		
Housing Status					25.176 ***	0
Private housing	2568	46.6	2946	53.4		
Rented housing	710	39.8	1075	60.2		
Health Status					53.036 ***	0
Good	2595	46.7	2964	53.3		
Fairly good	664	40.8	965	59.2		
Poor	19	17.1	92	82.9		
COVID-19 Status					320.934 ***	0
Prior COVID-19 infection	1061	64.2	592	35.8		
No prior COVID-19 infection	2217	39.3	3429	60.7		
Chronic Disease Status					11.736 **	0.001
Yes	175	54.2	148	45.8		
No	3103	44.5	3873	55.5		
Congenital Disease Status					54.061 ***	0
Yes						
No	218	64.3	121	35.7		
	3060	44	3900	56		
COVID-19 Vaccine Information Resources					429.541 ***	0
Electronic						
Non-electronic	2773	52.3	2527	47.7		
	505	25.3	1494	74.7		
Parents’ Education Level					514.483 ***	0
Elementary School	206	32.5	428	67.5		
Junior High School	294	23.3	968	76.7		
Senior High School	1382	44.4	1728	55.6		
University	1396	60.9	897	39.1		
Household Income					196.290 ***	0
<IDR 5,100,000	2875	42.6	3876	57.4		
≥IDR 5,100,000	403	73.5	145	26.5		
History of Family Members or Friends Infected with COVID-19					971.529 ***	0
Yes						
No	1496	74.3	517	25.7		
	1782	33.7	3504	66.3		
Parents’ Employment Status					201.209 ***	0
Employed	2958	48.6	3129	51.4		
Not employed	320	26.4	892	73.6		
Adolescents’ Knowledge Level of COVID-19 Vaccine					2965.669 ***	0
High	2824	76.1	888	23.9		
Low	454	12.7	3133	87.3		
Attitude Towards COVID-19 Vaccine					3562.956 ***	0
Positive	3053	76.6	933	23.4		
Negative	225	6.8	3088	93.2		
Confidence in COVID-19 Vaccine					3066.187 ***	0
High	2800	77.5	815	22.5		
Low	478	13	3206	87		

Note: ** *p*-value < 0.01: *** *p*-value < 0.001.

**Table 3 vaccines-10-01489-t003:** Factors associated with adolescents’ adherence to COVID-19 vaccination using binary logistic regression (*n* = 7299).

Variables	AOR	*p*-Value	95% CI	
			Lower	Upper
Willing to Get Vaccinated.				
Unwilling	0.159 ***	0	0.13	0.195
Doubtful	0.709 **	0.001	0.58	0.866
Ready	Ref			
Island				
Java	1.994 ***	0	1.6	2.485
Sumatra	1.571	0.066	0.97	2.545
Sulawesi	1.077	0.525	0.856	1.356
Borneo	1.284	0.419	0.7	2.355
Bali/East and West Nusa Tenggara	5.168 **	0.009	1.52	17.569
Maluku and Papua	Ref			
Age				
Early adolescent (12–14 years)	0.498 ***	0	0.417	0.595
Middle adolescent (15–17 years)	Ref			
Sex				
Male	0.997	0.972	0.849	1.171
Female	Ref			
Residential Area				
Urban	2.007 ***	0	1.701	2.368
Rural	Ref			
Distance from House to COVID-19 Vaccination Site				
<4 km				
>4 km	0.974	0.729	0.839	1.131
	Ref			
Housing Status				
Private housing	0.555 ***	0	0.46	0.67
Rented housing	Ref			
Health Status				
Good	3.273 **	0.001	1.651	6.488
Pretty good	2.030 *	0.045	1.016	4.056
Poor	Ref			
COVID-19 Status				
Prior COVID-19 infection	0.985	0.877	0.812	1.194
No prior COVID-19 infection	Ref			
Chronic Disease Status				
Yes	0.849	0.356	0.601	1.201
No	Ref			
Congenital Disease Status				
Yes	1.348	0.101	0.943	1.926
No	Ref			
COVID-19 Vaccine Information Resources				
Electronic				
Non-electronic	0.918	0.369	0.763	1.106
	Ref			
Parents’ Education Level				
Elementary School	0.857	0.311	0.635	1.155
Junior High School	0.396 ***	0	0.312	0.502
Senior High School	0.796 *	0.011	0.667	0.949
University	Ref			
Household Income				
<IDR 5100000	0.722 *	0.036	0.532	0.979
≥IDR 5100000	Ref			
History of Family Members or Friends Infected with COVID-19				
Yes				
No	1.345 **	0.002	1.116	1.622
	Ref			
Parents’ Employment Status				
Employed	1.287 *	0.02	1.04	1.592
Not employed	Ref			
Adolescents’ Knowledge Level of COVID-19 Vaccine				
High	1.962 ***	0	1.568	2.455
Low	Ref			
Attitude Towards COVID-19 Vaccine				
Positive	7.072 ***	0	5.652	8.849
Negative	Ref			
Confidence in COVID-1 Vaccine				
High	1.872 ***	0	1.495	2.343
Low	Ref			

Note: ** p*-value < 0.05; ** *p*-value < 0.01: *** *p*-value < 0.001.

## Data Availability

All the data are available from the corresponding author up on a reasonable request.

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
