# Peer review of "The Role of Knowledge, Attitude, Confidence, and Sociodemographic Factors in COVID-19 Vaccination Adherence among Adolescents in Indonesia: A Nationwide Survey"

_vaccines, 2022, doi:10.3390/vaccines10091489_

Round 1

Reviewer 1 Report

Thank you for the opportunity to review the manuscript titled “The role of knowledge, attitudes, confidence, and sociodemographic factors in COVID-19 vaccination adherence among adolescents in Indonesia: a nationwide survey.”

This study on adherence is important in light of the problem of low youth vaccination rates for the COVID-19 vaccine. The strength of this study is that a nationwide questionnaire survey was conducted in Indonesia.

I would like to ask you to consider a few points.

1)     The author states that teenage vaccination rates in Indonesia are low, but I would appreciate specifics on the vaccination rates in Indonesia, including among young people.

2)     Please describe the infection status in Indonesia at a given point in time. Also, please consider whether regional differences in vaccination may also be related to regional differences in infection status.

3)     Most of those who did not adhere to vaccination cited "parental prohibition" as the reason and were in mid-adolescence. In general, adherence to parental prohibition seems to be more common in early than in mid-adolescence. Please discuss this point.

4)     The only mention of gender differences is that most of those who did not adhere to the vaccination were women. Please describe whether there are gender differences in other results.

Reviewer 2 Report

I read with great interest the manuscript from Efendi et al. This is  a cross-sectional study on adherence to COVID-19 vaccination in Indonesia. The manuscript is quite well written and methods are adequate. Results support conclusion that highlight the need to improve adherence to COVID-19 vaccination in adolescents. However, the paper lack of novelty (see MMWR Morb Mortal Wkly Rep. 2020 Aug 21;69(33):1109-1116,  Public Health Manag Pract. 2021 May-Jun 01;27(3):278-284, Vaccines (Basel). 2021 Apr 10;9(4):366 & J Adolesc Health. 2021 Dec;69(6):925-932). However, the study represents an insight into the vaccination status of Indonesian adolescents. Here are my suggestions:

- Introduction section is too long. I would suggest to focus the paragraph in no more than 4-5 sentences to not stray to far from the object of the study;

- In the methods section is not totally clear whether the questionnaires used are original or borrowed from other studies. The section is long and confused and it would be recommended to clarify this issue and synthetize as much as possible;

- table 2 is misleading. All the variables present statistical significance but is not clear the comparison the author want to propose. Please clarify;

- In table 3 is reported "AOR". I guess it stand for "adjusted odds ratio". Adjust for which variables? Please clarify;

- Discussion is also too long. I would suggest to focus this section on relevant variables to strengthen the message the authors want to give to the readers

Language revision is recommended.

Reviewer 3 Report

The paper concerns nationwide research regarding the Role of Knowledge, Attitude, Confidence, and Sociodemographic Factors in COVID-19 Vaccination Adherence Among Adolescents in Indonesia. 

Genaral Commenta

1.     The manuscript is poorly written and after some major revision could be suitable for publication. 

2.     Many syntactic errors and problems. There are numerous errors of "Tense and Grammar" throughout the manuscript. Therefore, a diligent editing is in order to fix the ENGLISH language.

  1. Poor literature review and the section of introduction must be updated and be rephrased. 

More particular:

1.          Line 52: “The majority of the studies showed that the adolescents' reluctance to get vaccinated was mostly due to worries about vaccine safety, the effectiveness of vaccines, and the potential side effects (Liu et al., 2022)”. 

The authors claim that the majority of the studies but they were referred to only one reference. Could you please explain?

2.     The Authors have failed to state clearly the aim of the study. Please add research hypothesis and aim of the study at the end of Introduction section.

3.      The Material and Method section and more particularly the 2.2 “Measures” is unnecessarily VERBOSE and the sentiments expressed in this are quite REDUNDANT. They can be reduced by HALF without losing the key points. Table with the questions or the questionnaire in supplemental material will be more informative.

4.     Line 91: Authors referred some reasons that based on that the individuals were excluded from the study. However, in line 192, the authors stated that “Adolescents who were not 193 vaccinated for nonmedical and technical reasons (e.g., perceptions related to the vaccine 194 and its effects, parental prohibition) were regarded as the disobedient group.”, this sentence cause misunderstandings. Please rephrase and clarify. 

5.     Line 204-207. Please refer the reasons for choosing these particular statistics tests. 

6.     Line 216-218: rephrase the sentence

7.     Results section lacks about why the authors had chosen these statistical tests to analyze the data. The statistical analysis method is also missing. Interested remains the fact that the majority of the variables appears statistically significant results. Are there any previous research papers that they confirm the referred results?

8.     Figure 1: represents the results between individuals that they were vaccinated with one or two dose. Hoever, in Line 91, Authors claimed that: “Adolescents who 91 had not taken the first and or second dose for the following reasons were excluded from 92 this study”. Please explain and rephrase is needed. 

Round 2

Reviewer 1 Report

The points I raised seemed to have been fully considered and corrected. I think it is a fine paper.

Reviewer 3 Report

The Authors take under consideration all the comments on improving the readability of the manuscript. I accept the manuscript's publication in present form.